

# Computerized monitoring of COVID-19 trials, studies and registries in ClinicalTrials.gov registry

Craig S. Mayer and Vojtech Huser

Lister Hill National Center for Biomedical Communication, National Library of Medicine, NIH, Bethesda, MD, USA

## ABSTRACT

Clinical trial registries can provide important information about relevant studies for a given condition to other researchers and the public. We developed a computerized informatics based approach to provide an overview and analysis of COVID-19 studies registered on ClinicalTrials.gov registry. Using the perspective of analyzing active or completed COVID-19 studies, we identified 401 interventional clinical trials, 287 observational studies and 64 registries. We analyzed features of each study type separately such as location, design, interventions and update history. Our results show that the United States had the most COVID-19 interventional trials, France had the most COVID-19 observational studies and France and the United States tied for the most COVID-19 registries on ClinicalTrials.gov. The majority of studies in all three study types had a single study site. For update history "Study Status" is the most updated information and we found that studies located in Canada (2.70 updates per study) and the United States (1.76 updates per study) update their studies more often than studies in any other country. Using normalization and mapping techniques, we identified Hydroxychloroquine (92 studies) as the most common drug intervention, while convalescent plasma (20 studies) is the most common biological intervention. The primary purpose of most interventional trials is for treatment with 298 studies (74.3%). For COVID-19 registries we found the most common proposed follow-up time is 1 year (15 studies). Of specific importance and interest is COVID-19 vaccine trials, of which 12 were identified. Our informatics based approach allows for constant monitoring and updating as well as multiple applications to other conditions and interests.

## INTRODUCTION

The purpose of clinical trial registries, among others, is to inform the research community about currently ongoing studies. A registry can also be a sole source of study results for thousands of studies that would otherwise not publish a result article in a journal (*Zarin et al., 2019*). For the current effort to address the COVID-19 epidemic, this function of registries is of great value. The fields of clinical informatics and clinical research informatics have an important role to play in fighting the epidemic (*Moore et al., 2020*).

Corresponding author
Craig S. Mayer,
craig.mayer2@nih.gov

We focus on a single registry, ClinicalTrials.gov (CTG), and analyze COVID-19 registered studies. We chose to focus on CTG because it collects a rich set of metadata, supports record updates (*Fleminger & Goldacre, 2018*), allows for basic summary results deposition and studies in CTG make up a large volume of all studies tracked by the World Health Organization registry (*Huser & Cimino, 2013*). The goal of our study is to demonstrate how automated and clinical research informatics (*Embi & Payne, 2009*) methods can be used to analyze a set of closely related studies, as well as to use general statistical principles to analyze and understand key metrics for studies on a given condition or in a specific clinical domain. The computer code written in R language is open source and available at the project repository.("regCOVID Project Repository," 2020; https://github.com/lhncbc/r-snippets-bmi/tree/master/regCOVID) Our project differs from past published analyses of COVID-19 clinical studies (*Rosa & Santos, 2020*; *Fragkou et al., 2020*; *Checcucci et al., 2020*) by not using manual review of study records and instead relying fully on study metadata recorded in the registry.

Unlike manual approaches, our approach of using automation to monitor COVID-19 studies allows for quick and efficient, continuous monitoring of the state of COVID-19 research. Automated queries can provide an instant overview of COVID-19 research. They also allow for computing and visualizing current COVID-19 research trends. Furthermore, the informatics approach we use improves the capability for effective modulation and allows individuals to specifically target the desired information wanted to inform their research decisions (such as on clinical guidelines, and the use of certain interventions). With the gradual publication of COVID-19 study results, this approach also allows for automated detection of registry deposition of basic summary results and the publication of a study result journal article that is clearly tied to a registered COVID-19 trial, study or registry.

## MATERIALS AND METHODS

Our study had two aspects. The first aspect was assuming a journalist perspective and the motivation was whether CTG registry can provide a useful overview of COVID-19 studies without any manual curation. We wanted to demonstrate on a COVID-19 use case whether existing study metadata currently collected by CTG registry are accurate and adequate for a journalist interested in a registry-based picture of COVID-19 research. The methods and results sections address mostly this first aspect.

The second aspect was assuming an informatics or data science perspective and the motivation for this was to apply additional rules, data transformations and heuristics to CTG metadata that could characterize the quality of the registry data and possibly narrow the list of all CTG studies to a smaller set. The critical component of this second informatics aspect was a vision to generalize the fully computerized single disease report to all diseases. The discussion section of this article addresses this second aspect.

### Set of analyzed studies

We used the Aggregate Analysis of ClincalTrials.gov (AACT), which is a relational database version of CTG data that is created by parsing the XML (Extensible Markup

Language) representation of each study (*AACT Team, 2020*). It is published and maintained by Duke University. AACT data is typically 4 days behind CTG in terms of content or changes, which we deemed as acceptable. We performed separate analyses of COVID-19 studies based on their CTG study type of (1) interventional trials, (2) observational studies, and (3) registry-based studies (we use the term registries) (*CTG Team, 2020a*).

We designed several inclusion criteria to focus only on COVID-19 studies in scope for our analysis. This consisted of first, creating a search strategy based on title or study keywords. We also looked at CTG study metadata to select studies with fields that we considered relevant based on the triggering of quality measures and the connection to the regulatory process.

In terms of keyword and title search strategy, we evaluated three search approaches. The first method involved a search for the presence of keywords in the official title of the study. The keywords used were, "covid", "sars-cov", "2019-ncov", and "coronavirus". The second method searched for the same keywords in the free text condition field. The third method found studies that had a Medical Subject Heading (MeSH) term for the study of "coronavirus infections". We limited our search for each method to only include studies first submitted after 27 December 2019 (the date of the official report from Wuhan hospital to the local center for disease control and prevention). For later analysis, we present data for the first search method (with results for all three methods being available on the study repository at https://github.com/lhncbc/r-snippets-bmi/tree/master/regCOVID). The data presented below reflects the search performed on 11 May 2020. The results of the search methods were validated as appropriate COVID-19 studies via the manual review of the titles of a subset of the search results.

In terms of study inclusion criteria based on structured CTG's study metadata fields we used study status. Study status reflects the progress of the study from "not yet recruiting" to "recruiting" to "completed" (or other statuses such as "terminated" or "suspended") (*CTG Team, 2020a*). We elected to limit the scope of our analysis to reflect currently ongoing or completed COVID-19 studies. Given this assumption, we excluded studies with "not yet recruiting" study status. Under existing quality assurance rules used by CTG, studies in status "not yet recruiting" do not have to provide the study location (in terms of country) and we considered study country to be essential study metadata. Given our chosen analytical perspective of currently ongoing or completed COVID-19 studies, we excluded studies with an *unusual completion* status of "terminated", "suspended", or "withdrawn". However, we provide some results for unusually completed COVID-19 studies because when combined with the "Reason for termination" field, such studies may provide important insights.

## Analysis

For all study types, we analyzed a set of study metadata described below. Study metadata specific to a given study type (e.g., those collected only for observational studies or registries) are described in subsequent sections.

### Number of studies over time

Using the date when the study was first registered on CTG, we counted the number of total studies for each study type on a given date. We created a plot showing the temporal trend over time.

### Study sites and country

Despite the existence of national registries, CTG registry contains studies from many countries. For example, as of 2 June 2020, 61% of recruiting studies were located solely outside the US (according to CTG's overview page ("Trends, Charts, and Maps—ClinicalTrials.gov", https://clinicaltrials.gov/ct2/resources/trends)). Additional incentive for registration on CTG are FDA rules that require studies submitted in support of new drug applications to FDA to be registered at CTG. Similar requirements of CTG registration also exist from many study funders such as NIH, and journals such as International Committee of Medical Journal Editors (ICMJE) member journals.

We analyzed the geographic data for each study by reviewing the location fields in the CTG study record and counted the number of sites and identified the country of their locations. Each study could consequently include one or multiple countries.

### Study update activity

The clinical trial registry allows principal investigators to update the public about study completion, the final number of enrolled participants and basic summary results. High public interest in updates about COVID-19 studies was the main motivation for measuring update activity and study record recency.

We quantified the level of update activity for a study by looking at the number of updates and what fields are updated for a given study after its initial registration. We attempted to classify type of updates into technical updates (e.g., study sites changes) and updates of significant public interest (actual primary completion date or deposition of study results). We also evaluated the recency of the CTG study record by evaluating the number of days between the last update and the current date. Study update data is not available via AACT or the CTG Application Protocol Interface (API) (*CTG Team, 2020b*). However, information about study updates is available via the CTG website. To obtain this information we wrote an R script to scrape the data into a computable form.

We also evaluated the level of update activity for studies based in each country and found studies from which countries were more active in updating CTG study records.

### Study design

We analyzed CTG metadata pertaining to study design to classify studies. One feature analyzed for all study types was study enrollment (number of participants). CTG allows the reporting of estimated and actual enrollment into the trial. Study record managers can use this mechanism to publicly post updates about the number of enrolled participants.

## Study type specific analysis
### Interventional trials
*Interventional trial specific features*
We analyzed certain features which are specific to interventional trials alone. This includes phase, primary purpose, and number and type of arms.

*Intervention type*
For interventional trials specifically we analyzed the intervention types for the collection of COVID-19 studies. To do this we used CTG's metadata field of intervention type. CTG classifies each intervention as drug, device, biological, procedure, radiation, genetic, dietary supplement, behavioral, combination product, diagnostic test, and other. We counted the number of studies associated with each intervention type. Each study could include one or multiple intervention types. If multiple intervention types were included, we counted each study based on the combination of intervention types associated with the study. For example, NCT04334512, a "Study of Quintuple Therapy to Treat COVID-19 Infection", included two interventions of type "Drug" (Hydroxychloroquine and Azithromycin) and three interventions of type "Dietary supplement" (Vitamin C, Vitamin D and Zinc). This study was counted exactly once under a composite intervention type that consisted of the alphabetically sorted combination of two types: "dietary supplement|drug", with the "|" representing the term "and".

Our analysis of intervention types and names revealed that placebo as an intervention name is often used and captured under type "Drug" or "Biological". CTG type classification does not include placebo as a separate intervention type, however, we decided to experimentally create it and assign it based on a rule that looked for the term placebo in the intervention name.

*Interventions*
Researchers and the public are most interested to see which drugs (or other interventions) are being tested in relation to COVID-19. CTG allows study record administrators to specify intervention using free text and further assign interventions to study arms. Because of the vital importance of interventions and the correct counting of studies using the same intervention, we did implement a limited computerized method of processing free text interventions to achieve some semantic harmonization. After free text string transformations into harmonized intervention terms, we counted the number of studies that included a given intervention. We also evaluated the temporal change in the amount of studies for the most common interventions by showing the number of new studies on a weekly scale (as seen in a figure in the "Results" section).

From prior studies, there is an obvious need to harmonize semantically different interventions expressed as free text across different studies (*Cepeda, Lobanov & Berlin, 2013*). For example, the intervention term "ruxolitinib" can semantically harmonize entries of "Ruxolitinib Oral Tablet" (in study NCT04334044), "Ruxolitinib 10 MG" (NCT04338958) and "Ruxolitinib" (NCT04331665). Initial normalization involved the removal of extra white space and the conversion of each term to lower case. Representing

drug dose form was out of scope so further normalization removed commonly occurring dose form terms, such as, "tablet", "injection", and "pill".

Studies with multiple interventions were counted multiple times under each individual intervention. In some cases, the free text string for a single intervention (in the CTG data entry field) specified a combination of several interventions. Our transformation approach in such cases kept the combination as well as expanded the single entry into multiple separate interventions and counted each intervention separately. For example, NCT04334928 has an intervention that includes a combination of Emtricitabine and Tenofovir Disoproxil. In this case the study is counted once for Emtricitabine, once for Tenofovir, and once for the combination of Emtricitabine and Tenofovir. In some cases, the manual mapping reduced the term granularity and used a higher-level term.

*COVID-19 vaccine trials*

A segment of COVID-19 interventional trials with high importance and significant public interest are vaccine trials. CTG maintains a hierarchy of intervention types but vaccine as an intervention does not have a designated intervention type and is subsumed under the intervention type "Biological". Because there is no special vaccine intervention type, our method for finding vaccine trials was based on a string search for the term "vaccine" in the official title of the study. Once we created a COVID-19 vaccine trials subset, we applied on this set the same series of analyses and metrics mentioned above.

### Observational studies and registries

Observational studies and registries have metadata features that are not recorded for interventional trials. Such analyzed features were: time perspective and observational model for the set of observational studies and registries. In addition, one feature recorded and analyzed for only registries was follow up time.

## RESULTS

We developed a methodology to search and extract metadata on COVID-19 clinical studies. The database is a subset of the AACT database of ClinicalTrials.gov data. The database and result files can be found in our github repository at https://github.com/lhncbc/r-snippets-bmi/tree/master/regCOVID. The repository includes the R code (https://github.com/lhncbc/r-snippets-bmi/blob/master/regCOVID/regCovid_code_for_analysis.R), with comments explaining how it works, to obtain and analyze the data, as well as all comma separated value (CSV) data files used during the analysis. The repository also includes additional result data files not included in this article but described in the repository documentation. The repository also includes a list of descriptions for each data file (https://lhncbc.github.io/r-snippets-bmi/regCOVID/regCOVID_data_file_descript.html) for easy use. For example, the files, regCovid_all_studies-a.csv, regCovid_int-a.csv, regCovid_obs-a.csv, and regCovid_registry-a.csv are the lists of all studies, interventional trials, observational studies, and registries generated from search method A respectively. These files include all 64 columns from the AACT studies table, such as NCT ID, official title, start date, primary completion data, and enrollment. The description file has

more than 80 entries and provides guidance and descriptions for each included file in the analysis. Also included in the repository is an example of part of the code used in the analysis (https://github.com/lhncbc/r-snippets-bmi/blob/master/regCOVID/regCovid_example.R) and a quick-start tutorial (https://github.com/lhncbc/r-snippets-bmi/blob/master/regCOVID/regCovid_Tutorial.md) that shows users how to easily access and use our code and load the data files into R to review our results and perform their own analysis.

While this paper includes results from the main analysis done on 11 May 2020, the repository report is updated weekly and offers up to date results.

## Set of analyzed studies
### Search strategy
We found that the first search method, using the official title of the study, was the most comprehensive and included the most COVID-19 studies. The numbers listed below reflect only the search strategy and not applying the criteria based on study status.
As of 11 May 2020, the first search method returned a total of 1,302 studies. The second search method, based on the free text condition field, found fewer records (1,165 studies). The third method based on the MeSH term, returned 328 studies. The significant difference in the studies captured in the third search strategy is likely due to the fact, that there is no specific MeSH term for COVID-19 at this point and the MeSH condition field is not required and is left blank for many studies (38.2% of studies captured in the first search method left MeSH condition term blank).

We then applied metadata inclusion criteria (studies that are active, recruiting or completed and are not expanded access). This reduced the set for the first search method to 752 studies, the set from the second search method to 680 studies, and the set from the third search method to 210 studies.

This led us to select the set of COVID-19 studies generated from the first, most comprehensive search method, based on study title.

### Final study set
In terms of completion and presence of results, 48 studies in the final set were completed at the time of this analysis. None have provided summary results to this point. It is important to note that studies are typically required to submit results within 1 year after the primary completion date ("FDAAA 801 and the Final Rule—ClinicalTrials.gov", https://clinicaltrials.gov/ct2/manage-recs/fdaaa). Also, at the time of the analysis, 106 studies have past their primary completion date (12 studies when using primary completion day + 30 days) declared in the latest study record and have a status that indicates the study is still ongoing. This indicates that the record is possibly not kept current. Administrators do typically have 30 days after a status change to update the record (see 42 Code of Federal Regulation [CFR] 11.64(a)(1)(ii)) ("Frequently Asked Questions—ClinicalTrials.gov", https://clinicaltrials.gov/ct2/manage-recs/faq#updatesToCT). In an extreme case, 20 studies of those 106 studies have a status of "not yet recruiting" and are past their primary completion date.
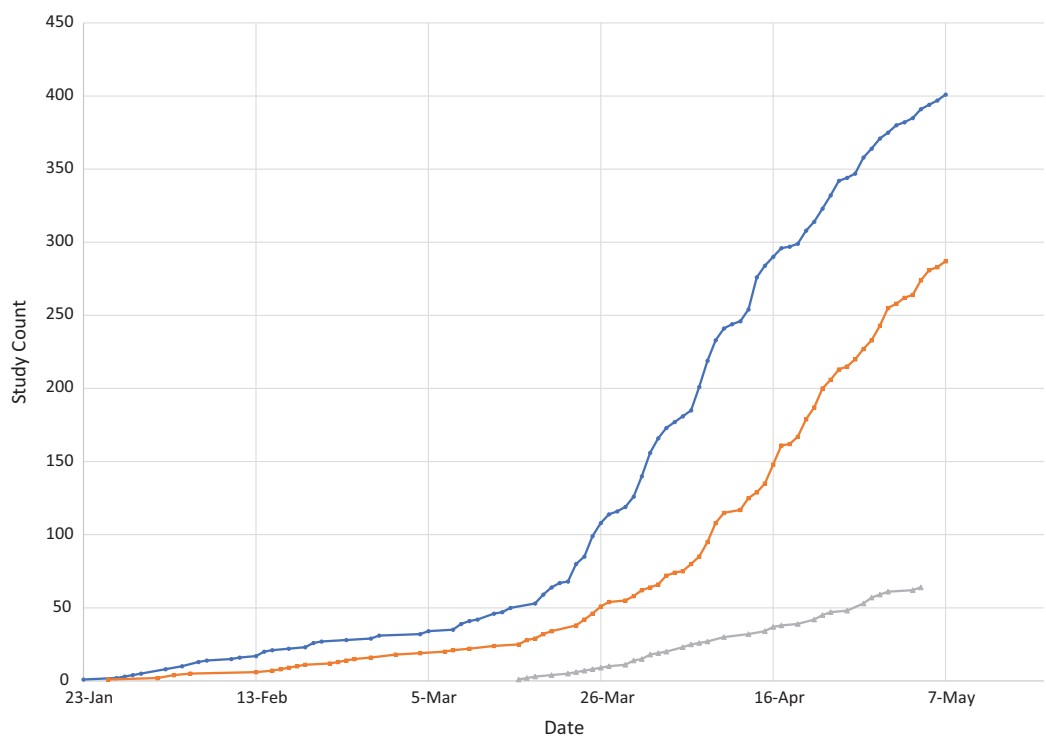

**Figure 1 Studies over time by study type.**

To understand how our metadata study inclusion criteria affects the final set, we briefly analyzed the set of studies excluded due to our study metadata criteria. The studies removed due to metadata included 516 studies that were not yet recruiting, 10 that were withdrawn, five that were suspended and two that were terminated. The reasons for termination of the two studies were "We cannot meet number of subjects as recently published similar studies" for NCT04357535 and "The epidemic of COVID-19 has been controlled well in China, no eligible patients can be enrolled at present" for NCT04257656. The interventions of the terminated studies were ACE-I (angiotensin-converting enzyme inhibitors) and ARB (angiotensin receptor blocker) for the former and Remdesivir for the latter. Our study type criteria also excluded 17 studies with a study type of "Expanded access".

## Studies over time

Figure 1 shows the number of registered studies over time by study type. Interventional trials are most numerous. An important regulatory consideration is that, in the US, applicable interventional clinical trials are required to register, while registration of observational studies and registries is optional. When considering the submission date, the first interventional trial, and the first study overall, was submitted to CTG on 23 January 2020, while the first observational study was submitted on 26 January 2020 and the first registry was not submitted until 12 March 2020.

**Table 1 List of countries where COVID-19 interventional trials are conducted.**

| Country | Study count | Percentage |
|---|---|---|
| United States | 121 | 30.2 |
| China | 49 | 12.2 |
| France | 42 | 10.5 |
| Spain | 23 | 5.7 |
| Italy | 19 | 4.7 |
| Brazil | 10 | 2.5 |
| Canada | 10 | 2.5 |
| Islamic Republic of Iran | 10 | 2.5 |
| Germany | 8 | 2.0 |
| Mexico | 8 | 2.00 |

## Analysis by study type

### Interventional trials

We identified a total of 401 COVID-19 interventional trials from CTG. These 401 studies included a total of 1,666 interventions.

*Study sites and country*

The majority of interventional trials had just one study site (259 studies, 64.6%%). A total of 41 studies had two sites and 18 studies had three sites, the second and third highest study counts. The study with the most sites was NCT04292730 ("Study to Evaluate the Safety and Antiviral Activity of Remdesivir in Participants With Moderate Coronavirus Disease (COVID-19) Compared to Standard of Care Treatment") with 183 sites.

As for country of operation, Table 1 shows the count of interventional trials by the country or countries that have at least one site that is part of the study.

The vast majority of studies 385 (96.0%) only included sites in a single country. Table 1 results indicate that the most common country for interventional trials was the United States with 121 studies (30.2%) followed by China with 49 studies (12.2%).

*Update activity*

We evaluated the amount of interventional trials that had updates after the study was first submitted to CTG (full update data are available in a report and as Comma Separated Value (CSV) files in the study github repository) ("regCOVID," 2020, https://lhncbc. github.io/r-snippets-bmi/regCOVID/regCovid_notebook2.html). At the time of the analysis, 71.1% (285 studies) of the 401 interventional trials show the presence of at least one update since first being submitted to CTG. The study with the most updates was NCT04280705, "Adaptive COVID-19 Treatment Trial (ACTT)" with 18 updates. The most common public interest and overall feature updated for COVID-19 interventional trials was "Study Status", which was updated 643 times including at least once by each of the 285 studies that had at least one update. Other commonly updated public interest fields include "Recruitment Status" (212 updates from 199 studies) and "Outcome Measures" (137 updates from 108 studies). The second most commonly

**Table 2 Number of updates per study by country (for countries with at least 8 COVID-19 interventional trials).**

| Country | Total updates | Study count | Changes per study |
|---|---|---|---|
| Canada | 27 | 10 | 2.70 |
| United States | 213 | 121 | 1.76 |
| Germany | 14 | 8 | 1.75 |
| Brazil | 17 | 10 | 1.70 |
| Spain | 38 | 23 | 1.65 |
| China | 65 | 49 | 1.33 |
| France | 53 | 42 | 1.26 |
| Iran | 12 | 10 | 1.20 |
| Mexico | 7 | 8 | 0.88 |
| Italy | 13 | 19 | 0.68 |

**Table 3 Overview of studies by study phase and number of participants (study size).**

| Phase | Study count | Percentage | 1st Qu. | # of participants: median (IQR)* | 3rd Qu. |
|---|---|---|---|---|---|
| N/A | 111 | 27.7 | 49.5 | 120 | 330 |
| Early Phase 1 | 7 | 1.7 | 10 | 10 | 40 |
| Phase 1 | 17 | 4.2 | 20 | 40 | 54 |
| Phase 1/Phase 2 | 23 | 5.7 | 20 | 72 | 190 |
| Phase 2 | 108 | 26.9 | 60 | 145 | 273.75 |
| Phase 2/Phase 3 | 34 | 8.5 | 108 | 269.5 | 433.5 |
| Phase 3 | 74 | 18.5 | 245 | 500 | 1215 |
| Phase 4 | 27 | 6.7 | 83 | 200 | 450 |

Note:
* IQR is interquartile range (1st quartile [25th percentile] and 3rd quartile [75th percentile]).

updated field overall, and most common technical field, was "Contacts/Locations", which was updated 393 times by 223 studies. Using 11 May 2020 as the current date, we also looked at the amount of days since last update to evaluate how current the existing CTG record is and found that the average amount of days since the last update is 20.6 days for all COVID-19 interventional trials.

Table 2 shows the amount of updates by studies in each country and the ratio of the number of updates compared to the number of studies in a given country. The table is limited to countries with at least eight studies. The country with the highest update rate is Canada with 2.70 updates per study, followed by the United States with 1.76 updates per study.

*Study design and interventional trial specific features*
*Study phase and size:* Considering study phase and study size (or enrollment; number of participants), Table 3 shows the counts of studies and percentage by study phase, as well as study size indicators: 1st quartile, median, and 3rd quartile for the participants enrolled (either actual or anticipated) for the set of studies of each phase.

**Table 4 Primary purpose of COVID-19 interventional trials.**

| Primary purpose | Study count | Percentage |
|---|---|---|
| Treatment | 298 | 74.3 |
| Prevention | 41 | 10.2 |
| Other | 19 | 4.7 |
| Supportive Care | 17 | 4.2 |
| Diagnostic | 15 | 3.7 |
| Health Services Research | 7 | 1.7 |
| Basic Science | 2 | 0.5 |
| Screening | 2 | 0.5 |

**Table 5 Count of the number of arms by arm type.**

| Arm type | Arm count |
|---|---|
| Experimental | 489 |
| Active Comparator | 160 |
| Placebo Comparator | 118 |
| No Intervention | 87 |
| Other | 43 |
| Sham Comparator | 3 |

Table 3 shows that the phase with the most studies is N/A with 111 studies (27.7%), which represents studies of intervention type device or behavioral. The second most common phase is Phase 2 with 108 studies (26.9%). Unsurprisingly the phase with the highest enrollment is Phase 3 with a median of 500 participants, while the lowest enrollment is Early Phase 1 with a median enrollment of 10 participants.

*Arms:* Considering number of study arms, most interventional trials have two arms (245 studies, 61.1%), while 73 studies (18.2%) have just one arm.

*Primary purpose:* Considering study primary purpose, Table 4 presents the breakdown into 8 purpose categories. In 298 (74.3%) of the analyzed COVID-19 interventional trials, the primary purpose was treatment. For 41 (10.2%) the primary purpose was prevention.

*Arm type:* CTG allows study managers to specify the type of each study arm. Each study arm is named and is classified as a specified arm type. Each study could have one or multiple arms of the same type. For example, NCT04321993, "Treatment of Moderate to Severe Coronavirus Disease in Hospitalized Patients", has three arms of type "Experimental" and one arm of type "No Intervention". One arm in this study has patients receiving an intervention of Lopinavir/Ritonavir, the second has patients receiving an intervention of Hydroxychloroquine, and the third has patients receiving an intervention of Barictinib. This study also has a fourth arm of patients receiving no intervention.

Considering types of all COVID-19 interventional trials, we found that the most common arm type is "Experimental", which appears 489 times. Table 5 shows the complete data for arm type in the set of 401 interventional trials.

| Table 6 Count of intervention types included in interventional trials. | | |
|---|---|---|
| **Composite intervention type** | **Study count** | **Percentage** |
| Drug | 137 | 34.2 |
| Drug\|Placebo | 75 | 18.7 |
| Biological | 32 | 8.0 |
| Other | 31 | 7.7 |
| Device | 22 | 5.5 |
| Drug\|Other | 22 | 5.5 |
| Behavioral | 12 | 3.0 |
| Biological\|Placebo | 12 | 3.0 |
| Diagnostic Test | 10 | 2.5 |
| Procedure | 8 | 2.0 |
| All Other Types* | 40 | 10.0 |

Note:
* This row combines rare Composite Intervention Types, such as "Drug|Biological", "Dietary Supplement", or "Device|Procedure" (see repository report for full table of intervention types) ("r-snippets-bmi/regCOVID at master · lhncbc/r-snippets-bmi").

Different diseases at different maturity of clinical research may be employing a different design, such as the inclusion of a placebo or active comparator. We calculated the placebo index, which is the percentage of interventional trials that have a placebo or sham comparator arm. Each study can have one or multiple arms that are assigned a placebo comparator. For our set of COVID-19 studies, the placebo index was 28.7% (115 of 401 total trials). We also calculated the active comparator index, which is the percentage of trials with at least one active comparator arm, and found that 28.9% (116 trials) have at least one active comparator arm.

*Intervention type*
Table 6 shows the count of studies by intervention type. Intervention type "Drug" is the most common (137 studies [34.2%]). The combination of drug and placebo intervention type was the second most prevalent with 75 studies (18.7%). Biological was the next most prevalent type with 32 studies (8.0%). Based on our methodology for classifying intervention types, each study can be counted only under one composite intervention type.

*Interventions*
There were a total of 449 distinct interventions listed prior to the implementation of our normalization and mapping process. Once the interventions were mapped the amount of normalized interventions was reduced to 403. The full mapping is available at the study repository (file: intervention_map2.xlsx) ("regCOVID Project Repository," 2020; https://github.com/lhncbc/r-snippets-bmi/tree/master/regCOVID). Table 7 shows the most common interventions used in COVID-19 interventional trials. Given our counting methodology for interventions, each study can be counted multiple times in Table 7 because combined interventions are expanded into their components as well as kept as a combination. The most common drug intervention was Hydroxychloroquine with 92

**Table 7  Most frequent interventions by study count (with a minimum study count of 13).**

| Intervention type | Intervention | Study count | Date when first appeared |
|---|---|---|---|
| Placebo | Placebo | 99 | 20-Februay-2020 |
| Drug | Hydroxychloroquine | 92 | 06-February-2020 |
| Other | Standard care | 40 | 23-January-2020 |
| Drug | Azithromycin | 24 | 23-March-2020 |
| Drug | Ritonavir | 24 | 28-January-2020 |
| Drug | Tocilizumab | 21 | 09-March-2020 |
| Biological | Convalescent plasma | 20 | 23-March-2020 |
| Drug | Lopinavir | 20 | 28-January-2020 |
| Drug | Lopinavir/Ritonavir | 16 | 30-January-2020 |
| Drug | Chloroquine | 13 | 19-March-2020 |

studies, followed by Azithromycin with 24 studies. The two (Hydroxychloroquine and azithromycin) appeared together four times. The most common combination intervention was Lopinavir/Ritonavir with 16 studies. We also found the presence of interventions most likely listed as a comparator or a non-intervention group, rather than a specific intervention. This is seen as 99 studies have placebo listed as an intervention while another 40 studies have standard care listed.

As for non-drug interventions, the most common biological is convalescent plasma with 20 studies. Other leading interventions for different types (not shown in Table 7) include oxygen supplying equipment for device with six studies and Vitamin C for dietary supplements with four studies. We also found that the same intervention can be listed as different intervention types. For example, convalescent plasma was listed for 14 studies as the intervention type biological, three times as other and three times as drug. We combined each intervention to count as the most commonly used intervention type when counting the intervention. For this case of convalescent plasma, that would count as 20 studies and categorize convalescent plasma as having the type biological.

*Interventions over time:* We also evaluated the temporal change for the most common interventions by analyzing the amount of new studies weekly for the most common interventions as seen in Fig. 2.

The plot shows how most interventions, including the most common intervention of hydroxychloroquine, peak in new weekly studies in early April. The plot also shows the later emergence of other interventions, such as convalescent plasma (shown in green).

*COVID-19 vaccine interventional trials*
Our search method for vaccine trial intervention type studies identified 12 COVID-19 vaccine trials, that also met our inclusion criteria of being active, recruiting or completed. Due to their high significance and increased public interest, it is interesting to consider how frequently such trials are updated. A total of nine trials (75.0% of the 12 vaccine trials) have at least one update and the median amount of updates is two. Considering the study country, six different countries have at least one vaccine trial, with China

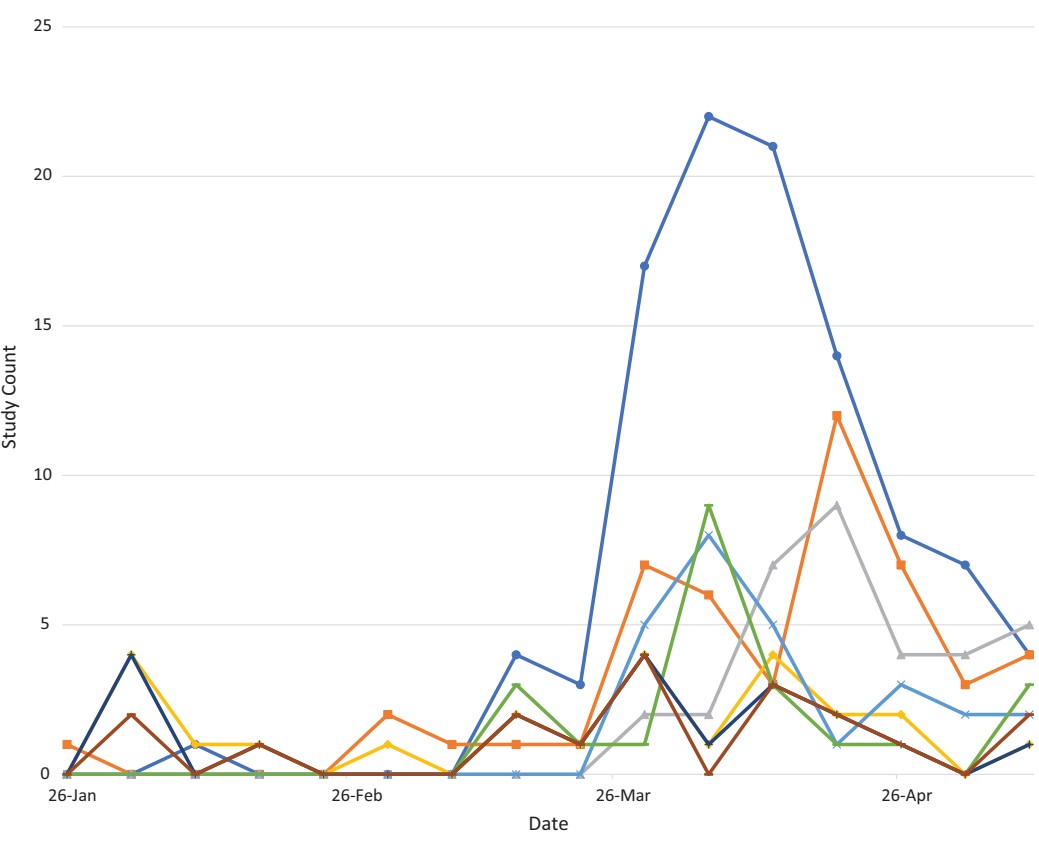

**Figure 2  Plot of new studies weekly for selected frequent COVID-19 interventions over time.**

(five vaccine trials) having the most, followed by the US with three trials. Five of the trials were Phase 1, six were Phase 1/Phase 2 and one was Phase 2. Of note is the fact that Phase 1 trials are not "applicable clinical trials" (as defined in US regulations) and such trials have no mandatory registration ("FDAAA 801 and the Final Rule—ClinicalTrials.gov", https://clinicaltrials.gov/ct2/manage-recs/fdaaa). Exactly half of vaccine interventional trials (six trials) had more than one site. As for design, the average number of arms was 5.4 with a median overall trial enrollment of 119.5 participants. The 12 vaccine interventional trials also included 52 experimental arms and seven placebo comparator arms. The full overview of all metadata parameters for vaccine trials (as well as for observational studies and registries described in subsequent sections) is available at the study repository ("regCOVID," 2020, https://lhncbc.github.io/r-snippets-bmi/regCOVID/regCovid_notebook2.html).

## Observational studies

We found a total of 287 observational studies. Similarly, to interventional trials, the vast majority of observational studies had just one site (238 studies, 82.9%). The country with the most observational studies was France with 75 (26.1%), followed by the United States with 47 (16.4%). Observational studies are updated less frequently than interventional trials as only 52.6% (151 studies) of the COVID-19 observational studies

have been updated since first being submitted to CTG (compared to the 71.1% of interventional trials that have been updated at least once). The observational study with the most updates was NCT04334954 "SARS-COV2 Pandemic Serosurvey and Blood Sampling" with 25 updates since registration on 6 April 2020. The most commonly updated public interest feature for observational studies was the "Study Status" which was updated 270 times by 151 studies and the most common technical feature updated was "Contacts/Locations" with 99 updates from 83 studies.

The median enrollment for observational studies was 353 participants. One feature of observational study design is the time perspective. A majority of the observational studies analyzed were prospective (180 studies, 62.7%), as opposed to 58 studies (20.2%) which were retrospective. For observational model, 167 of the observational studies (58.2%) use a cohort model. The second most commonly used model for the analyzed observational studies was case (45 studies, 15.7%).

Contrary to our expectation, we found observational studies that included interventions in their CTG record. Of the 287 observational studies, 179 (62.4%) listed something in the free-text intervention field. However, this number is misleading as in many cases the listed intervention was something that stated that there was no intervention (such as "no intervention", "observation", "non-interventional", etc.). Of the listed interventions most are listed as intervention type "Other" (86 studies, 30.0%) or "Diagnostic Test" (34, 11.9%).

### Registries

We analyzed a total of 64 COVID-19 registries (shorter term for registry-based studies). Of these registries 52 (81.3%) were limited to one site. The largest number of sites was 53. The countries with the most COVID-19 registries were France and the United States with nine studies each. Similar to observational studies, just over half of the analyzed registries, 51.6% (33 registries), have been updated at least once since their first registration. Also similar to observational studies, the most common public interest update for registries is to the study status, which has been updated 56 times by all 33 registries with an update, and the most common technical update is to the contacts and locations with 28 updates from 18 studies.

The median enrollment for the set of registries was 388 participants. Registries have many specific design features that differentiate them from other study types. One is the presence of a targeted follow-up time. The most common follow-up time for the analyzed registries was 1 year for 15 studies (23.4%), which was listed as either "1 year" or "12 months" and was combined to get the accurate value. The shortest follow-up time was 1 day for NCT04331171, "Epidemiological Observation From a Smartphone Self-monitoring Application for Suspected COVID-19 Patients' Triage", while the longest targeted follow-up duration for a registry was 20 years, for NCT04359602, "COVID-19 Recovered Volunteer Research Participant Pool Registry". For registries, CTG collects their observational model (similar to observational studies). The majority of registries, 48 (75.0%), use a cohort model. Also similar to observational studies, registries

can include a time perspective. However, unlike observational studies, no registries are retrospective. Instead the time perspective is usually either prospective (50 studies, 78.1%), or cross-sectional (six studies, 9.4%). A cross-sectional perspective means that the observation or intervention is made at a single point in time rather than on a continuous or recurring basis (*CTG Team, 2020a*).

Like observational studies, more than half (53.1%, 34 of 64 registries) included an intervention in the free text field. These interventions also include many that are not representative of an actual intervention and rather state the absence of an intervention just like with the previously mentioned observational studies. This is also shown in the intervention type as 19 of the 34 registries (55.9%) have an intervention type of "other".

## DISCUSSION

Based on our two perspectives, we discuss separately COVID-19 studies results (journalist perspective) and data science implications (informatics perspective).

### COVID-19 studies

Our study developed a computerized approach of retrieving COVID-19 studies from CTG registry for analysis. CTG's study metadata facilitates the useful classification of studies into many relevant subgroups (e.g., by study design, size, phase, recruitment status or intervention). Availability of this data in a structured form (either via CTG's API or via structured XML or relational data files) provides analytical views that would be difficult or impossible to achieve without a registry. As of 11 May 2020 ( the date of primary analysis), no study had deposited basic summary results.

The results presented above were summarized as of 11 May 2020. Refreshed and more current data (released weekly) can be obtained at the project repository. ("regCOVID Project Repository," 2020, https://github.com/lhncbc/r-snippets-bmi/tree/master/regCOVID; "regCOVID," 2020, https://lhncbc.github.io/r-snippets-bmi/regCOVID/regCovid_notebook2.html). Weekly updated reports allow researchers, journalists or the general public to quickly obtain a snapshot of the ongoing COVID-19 research. For example, a weekly report intervention section (similar to Table 7) can reveal to many research teams concentrated on COVID-19 what interventions are being studied with what intensity. This analytical view would require tens of manual queries using the generic CTG web interface.

### *Study limitations*

Our study has several limitations. First, we only used CTG registry to look for COVID-19 studies. Within this registry, we evaluated three search strategies, but some relevant COVID-19 studies may possibly be missed. Without a benchmark gold standard of all COVID-19 studies, the recall of our search strategy cannot be evaluated. It was out of scope of this study to establish the precision of our search. Second, our semantic harmonization of interventions is based on manual mapping by a single expert. Third, there are significant limitations of the informatics-based approach compared to manual review.

### Related studies

For example, COVID-19 Evidence Service from Centre for Evidence-Based Medicine at University of Oxford offers more comprehensive reviews. It was out of scope of our project to offer results comparable to human review. *Fragkou et al. (2020)* used a search and manual review methodology to compile and analyze a set of COVID-19 interventional trials and their interventions. *Checcucci et al. (2020)* did a literature and clinical trial registries search based on built-in search criteria to review COVID-19 vaccine trials. *Rosa & Santos (2020)* did a manual search of CTG to analyze COVID-19 trials using repurposed interventions. Considering the existing published studies, we conclude that our study is the first study to rely solely on computerized data science methods to compile and analyze a set of COVID-19 interventional trials, observational studies and registries. Our approach of using computerized data science methods allows for the continuous monitoring of the current state of COVID-19 research with minimal additional effort compared to a resource intensive manual review methodology. During a continuously changing public health emergency, this ability for any researcher to quickly and efficiently monitor changes and trends in clinical research is invaluable in informing the direction of their research efforts.

## Data science perspective

During the creation of a fully computerized, disease-focused report about ongoing or completed clinical studies, we observed several informatics themes described below. Before we describe individual lessons learned, we want to re-emphasize how computable representation of clinical study metadata is a crucial enabler for creating disease-based research snapshots. Moreover, several features of ClinicalTrials.gov registry proved to be highly valuable for our project. Such features are: structured representation of study metadata (XML and relational database format), registry support for result deposition and record updates, and legal and funding source policy requirements to maintain accurate registry records. In our analysis, we were able to build on prior clinical informatics research projects. Our project also shows value in further developing clinical informatics methods for data and metadata representation, semantic harmonization through terminologies and standards. The following informatics lessons were learned:

*Updates:* Our study is the first to analyze the frequency of updates to a study in CTG. We believe that adding the ability to access study updates to the CTG's API would be a useful addition. Our results indicate that analyzing study update activity is helpful in distinguishing studies with possibly outdated metadata (e.g., studies in status "not yet recruiting" but are past their anticipated completion date with some grace period allowed for record updating). Our study is also the first to analyze update activity by country of study.

*Intervention (free text):* CTG collects intervention as free text and for some studies, provides a corresponding concept in Medical Subject Headings (MeSH) terminology. This intervention harmonization as MeSH concept is done post hoc rather than during study metadata entry by the study record manager. We found that the MeSH intervention

concept is present in less than half (47%) of COVID-19 analyzed studies. This analysis prompted us to develop the denormalization and mapping method that we used.

Another intervention-related observation is the difference in how intervention combinations are listed in the free text field. In some cases, the combination intervention (e.g., two drugs given to some study group in combination) is recorded as two separate entries and the group or arm free-text description provides a way to clarify the combined usage. In other case the same intervention combination is recorded together as a single entry. This dual way of recording combined interventions formed our methodology for the most comprehensive approach of counting interventions (count them as both combinations and as separate interventions). We did not analyze arm description and so we did not combine separated interventions, which may have been assigned to the same arm and used in combination. This may possibly lead to the undercounting of certain intervention combinations.

*Registries:* We find valuable that CTG currently allows registration of observational studies and registries. Designing a user interface for registration and study representation format that can accommodate various designs and studies is a challenging task. Due to specific characteristics of certain study types, further customization of user interface or additional data quality checks may further improve the registry value to many stakeholders. For example, registries do not typically post one-time study results and may not have the same concept of primary completion date. Instead, annual or other regular interval updates about number of participants and summary results for participant flow may be more applicable. Clarifications in the user interface for entering interventions for registries (and for observational studies) may prevent entries which declare a formally drug typed intervention with the title "no intervention".

### Generalizing report to other diseases

Our emphasis on fully computerized analysis of a COVID-19 set of studies was motivated by our larger vision to apply the R scripted report for all MeSH encoded diseases found within the CTG registry. We refer to this result as the regCTG project and report repository. regCTG allows analysis of research by MeSH keyword for all clinical domains. We generated reports for all MeSH terms with at least 100 registered studies. A collection of nearly 1,000 disease-based reports is available at https://github.com/lhncbc/CRI/tree/master/regCTG.

We consider this generalization from a COVID-19 research report to a research report for nearly 1,000 diseases an important result of our project.

In another follow-up research project for this COVID-19 case study, we have also built a disease-intervention snapshot knowledge base (called D-SHOT) that lists all interventions appearing in interventional trials for a given condition ("Project Repository for Disease Snapshot", https://github.com/lhncbc/r-snippets-bmi/tree/master/D-SHOT). This knowledge base of disease-intervention pairs has many parameters for each intervention, such as date when first introduced, count of regularly completed studies or count of unusually completed studies ("terminated", "suspended", or "withdrawn")

studying that intervention. Experience from semantic harmonization of CTG's free text field into terminology concepts gained during this COVID-19 project was crucial in these two follow-up projects by our team. A related, non-open source project called Sherlock, proprietary to Johnson and Johnson is similarly parsing CTG's terms into formal concepts (*Cepeda, Lobanov & Berlin, 2013*).

### Weekly results updates

While, the main analysis presented above was done on 11 May 2020 (main analysis date), thanks to the computed nature of the analysis, we have been producing weekly updated reports (available at the github repository). We have also been improving and adding to the automated report since the main analysis date based on the deposition of the first study results and the appearance of study results publications. As of the main analysis date, there were zero studies with results deposited on CTG. Because of data evolution during the article review and revision preparation, the latest weekly report on our github repository (as of 13 August 2020; update analysis date) now snows three interventional trials and one registry with results posted. Analysis of linked PubMed publications for completed interventional trials, found that of the 83 completed interventional trials at the point of secondary analysis, nine had linked PubMed publications (10.8%).

For the weekly reports and data in our github repository, we welcome change requests submitted by interested researchers. For researchers re-using our code and interested in making modifications, a free registration to access the AACT database is required (obtainable within hours).

## CONCLUSIONS

We developed a computerized, data science driven approach to monitoring COVID-19 interventional trials, observational studies and registries. We report on several metrics for the 401 interventional trials, 287 observational studies and 64 registries as of our analysis date on 11 May 2020. More current and weekly refreshed data is available at our github repository. We also demonstrated that our COVID-19 disease focused report can be generalized to all diseases represented within a clinical trial registry.

## ACKNOWLEDGEMENTS

The findings and conclusions in this article are those of the authors and do not necessarily represent the official position of NLM, NIH, or the Department of Health and Human Services. We would like to thank NLM CTG team for help and providing comments (Rebecca Williams) on earlier versions of this manuscript. We would like to thank Kin Wah Fung and Laritza Rodriguez for providing comments on drafts of this manuscript. We would like to thank AACT team from Duke University for producing relational database view of CTG data and all COVID-19 study record managers for time spent with study registration and updates (including using optional CTG metadata fields not required by US regulations).

### Funding

This research was supported by the Intramural Research Program of the National Institutes of Health (NIH)/National Library of Medicine (NLM)/Lister Hill National Center for Biomedical Communications (LHNCBC). The funders had no role in study design, data collection and analysis, decision to publish, or preparation of the manuscript.

### Grant Disclosures

The following grant information was disclosed by the authors:
National Institutes of Health (NIH).
National Library of Medicine (NLM).
Lister Hill National Center for Biomedical Communications (LHNCBC).

### Competing Interests

The authors declare that they have no competing interests.

### Author Contributions

- Craig S. Mayer conceived and designed the experiments, performed the experiments, analyzed the data, prepared figures and/or tables, authored or reviewed drafts of the paper, and approved the final draft.
- Vojtech Huser conceived and designed the experiments, performed the experiments, analyzed the data, authored or reviewed drafts of the paper, and approved the final draft.

### Data Availability

REGistered COVID-19 Interventional trials and observational Studies (regCOVID) data are available at GitHub: https://github.com/lhncbc/r-snippets-bmi/tree/master/regCOVID.

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
