# Peer review of "Computerized monitoring of COVID-19 trials, studies and registries in ClinicalTrials.gov registry"

_PeerJ, doi:10.7717/peerj.10261_

## Round 0.1 · original submission · Major Revisions

Your approach is interesting, however, there were multiple suggestions made by two reviewers. Please take them into account and modify your manuscript accordingly.

·

Basic reporting

The paper "Computerized monitoring of COVID-19 trials, studies
and registries in ClinicalTrials.gov registry" by Mayer and Huser summarizes and describes the distribution of more than 1000 COVID-19 studies and divides these studies into several subgroups (e.g. by study design, Country, intervention type, etc). During this study, authors laboriously assembled all available data and joined it in the github repository. However, access to the data is obstructed by poor code commenting.

Experimental design

The paper fully describes and classifies available COVID-19 studies. However, there are some questions and advice for authors to structures the work and make in to be much more profitable than simple study review:
-The comment on github "(will be updated frequently)" status is not enough for treating the study to be completed;
-Readme file lack of description in the 'code' field;
-The code file 'covid2_cm.R' should contain more commentaries.
-There should be an obviously called example (e.g. example.R) file to pull data from the repository;
-Also short quickstart tutorial should be helpful;
-In my opinion, the results section should start with an overall database description;
-The structure of the database is definitely lack study scheme and a listing of available data described with column definitions;

Validity of the findings

The approach designed by the authors is potentially groundbreaking since there are many laboratories concentrated on the COVID-19 problem and the way to quickly determine the specificity of the hypothesis is highly required. But to become so, the paper needs to be reworked to highlight it's perks. The available data is messy and hard-to-understand for someone who did not participate in the study.

Additional comments

Instead of data classification, it is probably better to concentrate efforts on the description of the database authors assembled, which will result in the excellent paper many researchers will find very useful.

Reviewer 2 ·

Basic reporting

The manuscript review is titled Computerized monitoring of COVID-19 trials, studies, and registries in ClinicalTrials.gov registry, submitted to PeerJ for review. The manuscript conforms to the expectations of the PeerJ guidelines, and the needs of the scientific community, especially during the time of this pandemic.

The authors of the manuscript have done an excellent job summarizing the context of the use of automation and informatics for repositories like Clinicaltrials.Gov (CTG) using COVID-19 and the ability to repurpose the methodology and design for other disease conditions. Appreciate the high quality and intensive research effort conducted by the authors. The content is very good but could benefit from some restructuring.

The level of interest from a methodology perspective is on par, but would greatly benefit from a detailed description of the need for data automation & informatics, specifically in the introduction/background section. Example https://www.ncbi.nlm.nih.gov/pmc/articles/PMC7216865/

Experimental design

The manuscript is within the scope of PeerJ’s area of medical sciences. The study design and methodology is well defined. The author described well the scope limitation and formulated well-defined methods for future studies. Additional comments in the general comment section.

Validity of the findings

The search strategy is very well written and has supporting raw data and codes available.
In the discussion section, the authors have made a good effort in categorizing the two facets to discussion journalist and data science perspective. The data science perspective lacks the description of the need for informatics. Although the authors mention the role of manual vs automation from past studies, would be helpful to acknowledge how automation of such datasets influences research in the area of public health emergencies, etc.

Additional comments

As a general comment, this manuscript is important and needs to be published. It is outlined like a methods manuscript and how those could be applied to specific studies, with appropriate analysis.

Major comments:
Row 78: Introduction - you suggest the goal of the study is to demonstrate automated and research informatics methods to analyze a set of closely related studies. I would suggest broadening this to highlight the application of general statistical principles. This would support the discussion in sections methods and results.

Recommendation on restructuring: The draft switches between what a reader would expect from row 108 2.1 Analyzed study section, between analysis of studies based on (1) interventional trials, (2) observational (3) registry studies, to section 2.2 Analysis, 2.3 Intervention Trials 2.4 Observational studies and registries. From a readership, perspective recommends having a section of 2.3 Study types and then sub-sections for interventional and, observational and registries.

Minor comments:
Row 65: recommend to add: what this kind of analysis will help inform? Just a broad general characterization such as clinical guidelines development, etc.

Row 82: COVID-19 clinical studiess , spelling error

Row 110: (3) Recommend modifying registry studies to registry-based studies

Row 113: ‘,’ to be removed before We

For figure 2, a higher resolution image might be helpful.

---

## Round 0.2 · Minor Revisions

Thank you very much for improving the manuscript. If you could clarify the concerns of the first reviewer, and show the links to the new files, it will improve the paper even more.

·

Basic reporting

The paper "Computerized monitoring of COVID-19 trials, studies
and registries in ClinicalTrials.gov registry" by Mayer and Huser was greatly improved after revision and consider most of the questions reviewers asked.

Experimental design

1. The discussion section of the manuscript was altered and revised with the huge updates in the Readme file, which is now much more clear than before the review;
2. regCovid_code_for_analysis.R and regCovid_example.R code now joins the required code for downloading and filtering the data from the AACT database;
3. The authors added a file called regCovid_example.R which store an obvious and well-commented example of the code;
4. The authors added regCovid_tutorial.md that contains manual for running the code along with HTML explaining content of the GitHub directory;

Validity of the findings

Even since authors greatly modified the results sections, it is still lack of the newly-added files e.g. regCOVID_data_file_descript.html, regCovid_tutorial.md and regCovid_example.R with links and short descriptions of the file content.

Additional comments

A huge perk to the understanding of the db structure should be considered a quick-start guide located at regCovid_tutorial.md, HTML with file descriptions located at https://lhncbc.github.io/r-snippets-bmi/regCOVID/regCOVID_data_file_descript.html and regCovid_example.R with example run.
However, I failed to find these links in the paper, which can harm the overall impression of the paper.
I would advise to aggregate response to reviewers and add them to the paper to make it more clear.

Reviewer 2 ·

Basic reporting

This is the revision to previously reviewed manuscript titled Computerized monitoring of COVID-19 trials, studies, and registries in ClinicalTrials.gov registry, submitted to PeerJ for review.
As a general comment, the overall structure of the revised manuscript addresses the questions or suggestions made to the authors. The introduction, methodology, analysis, and results are very well structured. The author's rebuttal to the major and minor suggestions have been answered well, and I do not have any further concerns from a structural or readership perspective.

Experimental design

The authors have made addition based on prior comments for CRI. This certainly helps with tying in relevance from an informatics perspective.

Validity of the findings

The results are supported by the access to raw data and R-code. The authors have made significant changes to the GitHub repository and provided extensive documentation with revised codes. The addition of weekly updates of data was found to be very useful and certainly adds high value to researchers for COVID-19 studies. No other concerns.

Additional comments

Overall, the manuscript is significantly improved with accepted responses to the questions from the previous review. The manuscript is very relevant to the current research needs and the scientific society would greatly benefit from this information. The manuscript conforms to the expectations of the PeerJ guidelines.

---

## Round 0.3 · accepted · Accept

Thank you for providing the requested links and references.